# Association between red blood cell distribution width-to-albumin ratio at admission and all-cause mortality in patients with acute pancreatitis based on the MIMIC-III database

**Qingsong Wu[1]☯*, Lianyi Liao[2]☯, Qingjun Deng[1]**

**1** Department of Critical Care Medicine, Chongqing Red Cross Hospital (People's Hospital of Jiangbei District), Chongqing, China, **2** Department of Scientific Research and Education, Chongqing Red Cross Hospital (People's Hospital of Jiangbei District), Chongqing, China

☯ These authors contributed equally to this work.

\* wuqingsong200912@sina.com

## Abstract

### Objective

The association between red blood cell distribution width-to-albumin (RDW/ALB) ratio (RAR) and all-cause mortality in patients with acute pancreatitis has not been fully delineated. The purpose of this study was to investigate the impact of RAR at admission on 28-day all-cause mortality in patients with acute pancreatitis.

### Design

This investigation was conducted as a retrospective analysis utilizing data from the Medical Information Mart for Intensive Care (MIMIC)-III database.

### Participants

Patients with acute pancreatitis were selected from the MIMIC-III database according to predefined eligibility criteria.

### Outcome

The outcome was the all-cause mortality rates within 28 days.

### Results

Upon screening and excluding ineligible participants, a total of 931 patients with acute pancreatitis who met the inclusion criteria were analyzed. The overall mortality at 28 days was 11.71%. The receiver operating characteristic (ROC) analysis indicated that RAR had a moderate predictive value for all-cause mortality at 28 days, with an area under the curve (AUC) of 0.669 (95%CI, 0.617–0.720; p<0.05), and the cutoff value was 4.39. Divide the

**Data Availability Statement:** In this study, data from public databases were analyzed. These data

can be found here: MIMIC-III: https://physionet.org/content/mimiciii/1.4/.

**Funding:** The author(s) received no specific funding for this work.

**Competing interests:** The authors have declared that no competing interests exist.

patients into a high RAR group and a low RAR group based on the cutoff value. Kaplan-Meier survival analysis demonstrated a statistically significant increase in 28-day mortality among patients in the high RAR group compared to those in the low RAR group. Multivariate analysis indicated that potassium levels, total bilirubin, blood urea nitrogen, lactate, partial thromboplastin time, neutrophil and RAR were independently associated with the 28-day mortality. Multivariate Cox regression analysis confirmed that an elevated RAR was independently associated with increased mortality at 28 day (HR, 2.72; 95% CI, 1.64–4.52; p < 0.001).

## Conclusions

This study demonstrated that RAR at admission functioned as a significant prognostic indicator for mortality in patients with acute pancreatitis.

## Introduction

Acute pancreatitis (AP) is a condition characterized by the inflammation of the pancreas, triggered by a variety of etiologies that result in acute tissue damage, including pancreatic edema, hemorrhage, and necrosis. The symptoms may range from abdominal pain, nausea, bloating, vomiting, and may even lead to acute multiple organ dysfunction. AP is the most common cause of gastrointestinal-related hospitalization in the United States, accounting for over two billion dollars in annual healthcare spending [1]. Despite advancements in the treatment of AP, it remains the second highest cause of total hospital stays, the largest contributor to aggregate costs, and the fifth leading cause of in-hospital deaths [2]. Identifying a simple, early, and relatively precise indicator for AP is critical for the implementation of effective interventions and for alleviating the substantial healthcare burden associated with this condition.

Current research has identified some biological markers related to the prognosis of AP, including the serum lactate [3], C-reactive protein to lymphocyte ratio [4], ferritin-to-hemoglobin ratio[5], ratio of red cell distribution width-to-total serum calcium [6]. As a component of the complete blood count test, RDW is routinely measured at admission for the vast majority of patients. It offers the benefits of being cost-effective and readily available, and it has emerged as a significant and reliable marker for both acute and chronic systemic inflammation [7].

Albumin is a crucial protein constituent in plasma, with primary roles that encompass sustaining plasma osmotic pressure, facilitating the transport and storage of nutrients, maintaining acid-base balance, immune regulation, antioxidant and anti-inflammatory effects. As a novel and convenient biomarker for inflammation, RAR is composed of the ratio of RDW to albumin, which has been proved to be valuable for the prognosis in patients with acute illnesses such as sepsis [8], chronic obstructive pulmonary disease exacerbation [9], acute kidney injury [10].

Nevertheless, the prognostic value of RAR in the patients with AP has not been thoroughly investigated. The current study aimed to explore the relationship between RAR and mortality in patients with AP through a retrospective analysis of a extensive database. The goal of this analysis was to pinpoint patients who are at a heightened risk for unfavorable outcomes, thereby facilitating targeted interventions and improved clinical management.

## Methods and materials

### Data source and permission for use

This study performed a retrospective analysis utilizing a large, publicly available database, the Medical Information Mart for Intensive Care-III(MIMIC-III). MIMIC-III comprises data related to 53,423 distinct hospital admissions for adult patients admitted to critical care units between 2001 and 2012. Data contains vital signs, medications, laboratory measurements, observations and notes charted by care providers, fluid balance, procedure codes, diagnostic codes, imaging reports, hospital length of stay, survival data, and more [11]. Dr. Qingsong Wu, an author of this study, concluded a mandatory training course, leading to permission for data extraction from the database for research purposes (certification number:44327064) on August 19, 2021. Since health information was anonymized, there was no access to information that could identify individual participants during or after data collection. The Massachusetts Institute of Technology and Beth Israel Deaconess Medical Center both examined and approved the human subjects-involved investigations. Patients' informed permission was not necessary for this study since health information was anonymized.

### Patients and data variables

Data extraction was performed by using Structured Query Language (SQL) programming within the PostgreSQL framework (version 14.0).

Patients diagnosed with AP were identified from database using the International Classification of Diseases, ninth revision (ICD-9, code 577.0). Exclusion criteria were as follows: patients under the age of 18, those with missing study data exceeding 20%, lack of data pertaining to RAR, and those with an intensive care unit (ICU) stay of less than 24 hours. For patients who had multiple ICU admission, only data from their first ICU stay were included in the analysis.

Following a meticulous screening of the eligible patient cohort, comprehensive and detailed assembly of baseline characteristics were undertaken to delineate the profile of patients suffering from AP. These parameters included a series of demographic details, comorbid conditions, vital sign records, laboratory indicators and interventions. Demographic details included age, gender, ethnicity. Comorbidity encompassed congestive heart failure, hypertension, chronic pulmonary, diabetes and renal failure. Vital sign records such as heart rate, blood pressure, respiratory rate and temperature were collected. Laboratory indicators such as potassium, total bilirubin (Tb), creatinine, blood urea nitrogen (Bun), lactate, hemoglobin (Hb), platelet, alanine aminotransferase (Alt), prothrombin time (PT), partial thromboplastin time (PTT), neutrophil (Neut), international normalized ratio (INR), hematocrit and glucose were recorded. Interventions included the use of mechanical ventilation. It is noteworthy that laboratory indicators were derived from the initial measurements taken within the first 24 hours of patient's admission to the ICU. Concurrently, the vital signs represented the mean values documented over the same initial 24-hour period. Furthermore, critical evaluation scores, including the Sequential Organ Failure Assessment (SOFA) and the Simplified Acute Physiology Score II (SAPSII), were gathered as part of the patient assessment. In addition, weight and urine output were also obtained. Urine output referred to the total volume of urine excreted within 24 hours after admission.

### RAR calculation and grouping

RAR was calculated by dividing the RDW (%) by the Albumin (g/L) concentration. Prior to stratifying the cohort, we performed ROC analysis to establish the optimal cutoff value, which was identified as 4.39. Subsequently, the cohort was categorized into two groups according to

this cutoff value. The groups were defined as follows: the low RAR group (2.51–4.39), and the high RAR group (4.39–13.29).

## Outcomes

The endpoint was to assess the all-cause mortality within 28 days. Mortality data were assured from the ICU admission records and register dates of death.

## Statistical analysis

In this study, variables with missing values exceeding the 20% threshold were excluded. The remaining variables with missing values were filled in through multiple imputation methods. This process was conducted using the mice package within the R statistical software, utilizing regression-based models for the imputation procedure.

To assess the presence of multicollinearity among variables, a collinearity evaluation was conducted prior to the main analysis. Categorical variables were exhibited as proportions (%), and the $\chi 2$ test was utilized to examine categorical variables. All continuous variables were represented as median and IQR on account of the Kolmogorov-Smirnov test which exhibited a non-normal distribution. The comparison of demographics and baseline characteristics for patients grouped by the RAR level was conducted. To reduce baseline imbalance between the two groups, propensity score matching (PSM) was also performed. PSM was applied with a caliper width of 0.1 logits of the standard deviation. The cohort was paired in a 1:1 ratio using the nearest neighbor matching technique. The efficacy of the PSM was evaluated by the standardized mean difference (SMD), with an SMD ≤0.1 indicating a balanced model with regard to initial characteristics. The Kaplan-Meier curve was employed to assess the survival probability at 28 days for the two groups of patients with different RAR levels. The effect of RAR on survival duration was dissected through univariate and multivariate Cox regression models. We established four models using multivariate Cox regression analysis, adjusting for different variables. Model 1 was unadjusted, serving as a baseline. Model 2 was adjusted for age, ethnicity, gender, comorbidities. Model 3 included additional adjustments for potassium, Tb, Bun, lactate, PTT and Neut levels. Model 4 further adjusted for mechanical ventilation. Moreover, we investigated the association between the RAR and mortality via restricted cubic splines with four knots at 5%, 35%, 65% and 95%. Furthermore, subgroups analysis was conducted to verify the consistency and robustness of our results. In our study, all statistical analyses were performed using R software (version 4.2.2), along with the use of SPSS software. A p-value<0.05 was considered statistically significant.

## Results

### Cohort composition

Fig 1 provided a detailed illustration of the inclusion and exclusion employed in this study. The research initiated with an original screening of 961 patients with AP to identify potential inclusion. A rigorous screening process was undertaken, adhering to the predefined inclusion criteria, which culminated in a final study cohort comprising 931 patients. Of these, 373 patients were classified into the low RAR group, while 558 patients were allocated to the high RAR group. After propensity score matching (PSM), 273 pairs of patients were matched.

### Baseline characteristics overview

The baseline characteristics table (Table 1) presented critical demographic and clinical variables grouped by cutoff value of RAR before and after PSM. Patients in the low RAR group

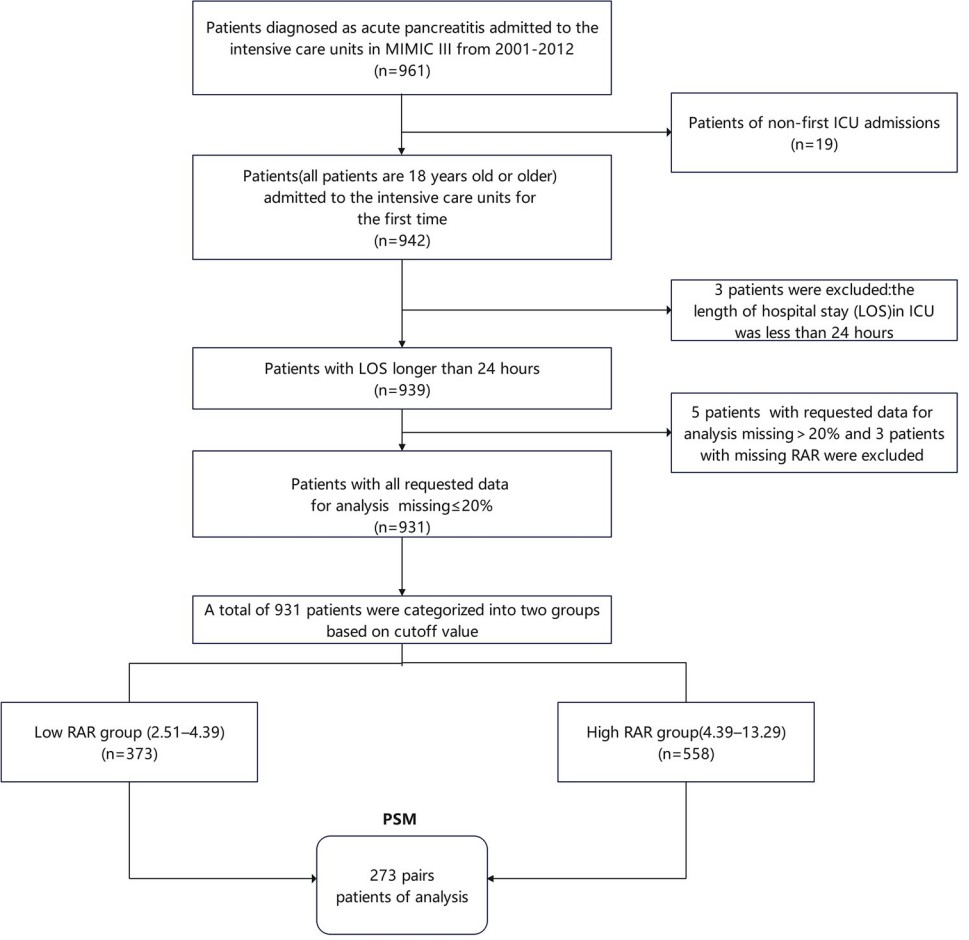

**Fig 1. Flow chart of the inclusion and exclusion procedure and PSM.** Abbreviation: LOS, length of stay; RAR, red blood cell distribution width-to-albumin ratio; ICU, intensive care unit; PSM, propensity score matching.

exhibited a higher SOFA score (median [IQR]: 5 [3–8]) compared to the survival group (4 [2–7]). In addition, the low RAR group presented with an increased respiratory rate at admission, as well as higher levels of lactate, glucose, hemoglobin, hematocrit, and ALT. Moreover, the high RAR group had higher levels of BUN, PT, PTT, and INR. No significant disparities were observed between the two groups in terms of comorbidities or the utilization of mechanical ventilation. Subsequently, we used propensity score matching to reduce the impact of confounding factors between the two groups, with a caliper value of 0.1. Ultimately, 273 pairs of AP patients were matched. There were no statistical differences in the baseline characteristics between the two groups after PSM. Furthermore, collinearity assessment was conducted by using variance inflation factor(VIF), the result showed that there were no significant overlaps among the variables under investigation (S1 Table).

## Study outcomes

The Kaplan-Meier survival analysis demonstrated a significantly higher 28-day mortality rate among patients in the high RAR group compared to those in the low RAR group, both in the unmatched cohort and the propensity score-matched cohort (Fig 2). Patients were stratified into groups based on their survival status at 28 days, and a comparison of baseline

**Table 1. Patient demographics and baseline characteristics before and after PSM.**

| Variables | Before PSM | | | | | After PSM | | | | |
|---|---|---|---|---|---|---|---|---|---|---|
| | Total(n = 931) | Low RAR group (n = 373) | High RAR group (n = 558) | p-value | SMD | Total(n = 931) | Low RAR group (n = 373) | High RAR group (n = 558) | p-value | SMD |
| Age (years) | 58.30 (46.75,70.56) | 58.19 (45.55,70.27) | 58.30 (47.31,70.77) | 0.287 | 0.074 | 58.30 (46.29,70.0) | 58.11 (46.05,70.03) | 58.30(46.93, 70.58) | 0.517 | 0.056 |
| Weight (kg) | 82.64 (72.75, 86.30) | 82.64(75.00, 85.00) | 82.64 (71.53,86.40) | 0.369 | -0.092 | 82.64 (75.00, 87.07) | 82.64 (76.30, 85.00) | 82.64 (73.90, 87.60) | 0.901 | 0.007 |
| SAPSII | 36.00 (26.00, 45.00) | 36.00 (25.00, 47.00) | 35.00 (26.25,45.00) | 0.568 | -0.062 | 35.00 (26.00, 45.00) | 35.00 (25.00, 44.00) | 36.00 (27.00, 45.00) | 0.179 | 0.112 |
| **SOFA** | 5.00 (3.00, 7.00) | 5.00 (3.00, 8.00) | 4.00 (2.00, 7.00) | 0.024 | -0.184 | 5.00 (3.00, 8.00) | 5.00 (3.00, 8.00) | 5.00 (3.00, 7.00) | 0.548 | 0.041 |
| Heart rate (times/min) | 94.00(81.01, 107.30) | 94.41 (80.97,107.47) | 93.72 (81.22,106.92) | 0.795 | -0.020 | 93.76(82.21, 107.63) | 93.62 (81.83,107.04) | 94.34(82.82, 108.03) | 0.790 | 0.021 |
| Bp (mm Hg) | 80.00 (72.48, 89.98) | 79.86 (72.90, 90.13) | 80.01 (72.36, 89.92) | 0.940 | -0.017 | 80.03 (73.01, 89.97) | 80.44 (73.88, 90.13) | 79.84 (72.36, 89.82) | 0.417 | -0.052 |
| **Respiratory rate** (times/min) | 20.29 (17.17, 23.30) | 20.68 (17.26, 24.06) | 19.97 (17.05, 22.88) | 0.049 | -0.140 | 20.23 (16.98, 23.38) | 20.63 (16.92, 23.88) | 19.90 (17.22, 22.91) | 0.232 | -0.095 |
| Temperature (˚C) | 37.03 (36.60, 37.49) | 37.02 (36.58, 37.47) | 37.05 (36.60, 37.50) | 0.898 | 0.011 | 37.02 (36.59, 37.48) | 37.00 (36.63, 37.48) | 37.05 (36.56, 37.49) | 0.766 | -0.008 |
| Urine output (ml) | 1642.00 (995.50,2530.00) | 1630.00 (940.00,2508.00) | 1667.50 (1021.00, 2530.00) | 0.558 | -0.018 | 1671.00 (1052.25, 2526.25) | 1695.00 (1140.00, 2755.00) | 1605.00 (996.00, 2370.00) | 0.124 | -0.190 |
| Potassium (mg/dL) | 4.15 (3.80, 4.15) | 4.15 (3.90, 4.15) | 4.15 (3.80, 4.20) | 0.617 | -0.016 | 4.15 (3.90, 4.15) | 4.15 (3.90, 4.15) | 4.15 (3.80, 4.15) | 0.549 | -0.063 |
| Tb (mg/dL) | 0.90 (0.50, 2.26) | 0.90 (0.50, 2.10) | 0.90 (0.50, 2.50) | 0.476 | 0.167 | 0.90 (0.50, 2.18) | 0.90 (0.50, 2.20) | 0.80 (0.50, 2.00) | 0.478 | -0.025 |
| Creatinine (mg/dL) | 1.10 (0.80, 1.85) | 1.10 (0.80, 1.70) | 1.10 (0.70, 2.10) | 0.508 | 0.163 | 1.10 (0.80, 1.80) | 1.10 (0.80, 1.70) | 1.10 (0.70, 1.80) | 0.663 | -0.012 |
| **Bun** (mg/dL) | 22.00 (14.00, 38.00) | 20.00 (13.00, 32.00) | 24.00 (15.00, 44.00) | < .001 | 0.207 | 21.00 (13.00, 33.00) | 20.00 (13.00, 32.00) | 22.00 (12.00, 35.00) | 0.586 | -0.033 |
| **Lactate** (mmol/L) | 2.10 (1.30, 2.66) | 2.50 (1.50, 2.66) | 1.95 (1.30, 2.69) | 0.013 | 0.027 | 2.20 (1.40, 2.70) | 2.40 (1.40, 2.66) | 2.10 (1.30, 2.90) | 0.749 | 0.057 |
| **Hb** (mg/dL) | 11.90 (10.30, 13.70) | 13.00 (11.70, 14.40) | 11.00 (9.70,12.67) | < .001 | -0.771 | 12.40 (11.03, 13.80) | 12.40 (11.10, 13.90) | 12.40 (11.00, 13.80) | 0.796 | -0.003 |
| **Glucose** (mg/dL) | 131.00 (104.00,175.00) | 140.00 (111.00,189.00) | 124.50 (99.00,164.75) | < .001 | -0.292 | 135.50 (106.00,183.75) | 137.00 (109.00,183.00) | 133.00 (105.00,184.00) | 0.391 | -0.040 |
| Platelet (10⁹/L) | 222.00 (157.00,310.00) | 230.00 (169.00,298.00) | 217.50 (149.00,319.2) | 0.163 | -0.060 | 224.50 (162.50,307.50) | 231.00 (166.00,308.00) | 221.00 (159.00, 300.00) | 0.428 | -0.144 |
| **Alt** (U/L) | 43.00(22.00, 133.50) | 55.00 (25.00,188.00) | 38.50 (20.25, 103.00) | < .001 | -0.093 | 48.00(22.00, 141.75) | 53.00 (24.00,145.00) | 46.00(21.00, 125.00) | 0.132 | -0.075 |
| **Hematocrit** (mm/h) | 30.40 (26.30, 34.55) | 32.30 (28.20, 36.10) | 29.10 (25.42, 33.18) | < .001 | -0.498 | 31.55 (28.00, 35.48) | 31.60 (27.60, 35.00) | 31.50 (28.10, 35.60) | 0.682 | 0.030 |
| **PT** (s) | 15.10 (13.55, 17.35) | 14.70 (13.30, 17.35) | 15.30 (13.80, 17.80) | < .001 | 0.095 | 14.95 (13.43, 17.35) | 14.70 (13.30, 17.35) | 15.10 (13.70, 17.35) | 0.149 | 0.029 |
| **PTT** (s) | 33.00 (27.80, 40.11) | 31.90 (26.90, 40.11) | 33.95 (28.30, 40.88) | 0.002 | 0.139 | 32.55 (27.52, 40.11) | 31.90 (26.70, 40.11) | 33.40 (28.30, 40.11) | 0.063 | 0.049 |
| Neut(10⁹/L) | 12.19 (7.81, 15.56) | 11.85 (7.63, 14.64) | 12.41 (7.85, 16.02) | 0.233 | 0.118 | 12.15 (7.99, 15.09) | 11.96 (7.93, 14.98) | 12.32 (8.06, 15.11) | 0.912 | 0.047 |
| **INR** | 1.40 (1.20, 1.75) | 1.30 (1.20, 1.75) | 1.40 (1.20, 1.75) | 0.003 | 0.084 | 1.40 (1.20, 1.75) | 1.30 (1.20, 1.75) | 1.40 (1.20, 1.75) | 0.252 | 0.012 |
| Ethnicity, n(%) | | | | 0.160 | | | | | 0.995 | |
| White | 622 (66.81) | 256 (68.63) | 366 (65.59) | | -0.064 | 360 (65.93) | 179 (65.57) | 181 (66.30) | | 0.015 |
| Black | 91 (9.77) | 31 (8.31) | 60 (10.75) | | 0.079 | 51 (9.34) | 27 (9.89) | 24 (8.79) | | -0.039 |
| Asian | 27 (2.9) | 11 (2.95) | 16 (2.87) | | -0.005 | 18 (3.3) | 9 (3.30) | 9 (3.30) | | 0.000 |
| Hispanic OR Latino | 32 (3.44) | 7 (1.88) | 25 (4.48) | | 0.126 | 14 (2.56) | 7 (2.56) | 7 (2.56) | | 0.000 |
| Other | 159 (17.08) | 68 (18.23) | 91 (16.31) | | -0.052 | 103 (18.86) | 51 (18.68) | 52 (19.05) | | 0.009 |
| Congestive heart failure, n (%) | | | | 0.736 | | | | | 0.762 | |

(*Continued*)

**Table 1.** (Continued)

| Variables | Before PSM | | | | | After PSM | | | | |
|---|---|---|---|---|---|---|---|---|---|---|
| | Total(n = 931) | Low RAR group (n = 373) | High RAR group (n = 558) | p-value | SMD | Total(n = 931) | Low RAR group (n = 373) | High RAR group (n = 558) | p-value | SMD |
| No | 711 (76.37) | 287 (76.94) | 424 (75.99) | | -0.022 | 417 (76.37) | 210 (76.92) | 207 (75.82) | | -0.026 |
| Yes | 220 (23.63) | 86 (23.06) | 134 (24.01) | | 0.022 | 129 (23.63) | 63 (23.08) | 66 (24.18) | | 0.026 |
| Hypertension, n (%) | | | | 0.397 | | | | | 0.171 | |
| No | 436 (46.83) | 181 (48.53) | 255 (45.70) | | -0.057 | 266 (48.72) | 141 (51.65) | 125 (45.79) | | -0.118 |
| Yes | 495 (53.17) | 192 (51.47) | 303 (54.30) | | 0.057 | 280 (51.28) | 132 (48.35) | 148 (54.21) | | 0.118 |
| Chronic pulmonary disease, n (%) | | | | 0.414 | | | | | 0.297 | |
| No | 780 (83.78) | 308 (82.57) | 472 (84.59) | | 0.056 | 457 (83.7) | 224 (82.05) | 233 (85.35) | | 0.093 |
| Yes | 151 (16.22) | 65 (17.43) | 86 (15.41) | | -0.056 | 89 (16.3) | 49 (17.95) | 40 (14.65) | | -0.093 |
| Diabetes, n (%) | | | | 0.530 | | | | | 0.301 | |
| No | 719 (77.23) | 292 (78.28) | 427 (76.52) | | -0.042 | 426 (78.02) | 218 (79.85) | 208 (76.19) | | -0.086 |
| Yes | 212 (22.77) | 81 (21.72) | 131 (23.48) | | 0.042 | 120 (21.98) | 55 (20.15) | 65 (23.81) | | 0.086 |
| Renal failure, n (%) | | | | 0.926 | | | | | 0.458 | |
| No | 800 (85.93) | 321 (86.06) | 479 (85.84) | | -0.006 | 470 (86.08) | 238 (87.18) | 232 (84.98) | | -0.062 |
| Yes | 131 (14.07) | 52 (13.94) | 79 (14.16) | | 0.006 | 76 (13.92) | 35 (12.82) | 41 (15.02) | | 0.062 |
| Mechanical ventilation, n (%) | | | | 0.864 | | | | | 0.122 | |
| No | 511 (54.89) | 206 (55.23) | 305 (54.66) | | -0.011 | 300 (54.95) | 159 (58.24) | 141 (51.65) | | -0.132 |
| Yes | 420 (45.11) | 167 (44.77) | 253 (45.34) | | 0.011 | 246 (45.05) | 114 (41.76) | 132 (48.35) | | 0.132 |
| Gender, n (%) | | | | 0.740 | | | | | 0.546 | |
| Male | 518 (55.64) | 210 (56.30) | 308 (55.20) | | -0.022 | 307 (56.23) | 150 (54.95) | 157 (57.51) | | 0.052 |
| Female | 413 (44.36) | 163 (43.70) | 250 (44.80) | | 0.022 | 239 (43.77) | 123 (45.05) | 116 (42.49) | | -0.052 |

Abbreviations: SAPSII, simplified acute physiology score II; SOFA, sequential organ failure assessment; Bp, blood pressure; Tb, total bilirubin; Bun, blood urea nitrogen; Hb, Hemoglobin; Alt, alanine aminotransferase; PT, prothrombin time; PTT, partial thromboplastin time; Neut, neutrophil; INR, International Normalized Ratio.

characteristics was conducted between the two groups (S2 Table). Before conducting COX regression analysis, the R software was used to perform the proportional hazards test through Schoenfeld residuals method. The results revealed that all independent variables met the proportional hazards assumption. Subsequently, both univariate and multivariate Cox regression analyses were conducted to identify risk factors for the 28-day mortality (S3 Table). The multivariate analysis indicated that potassium levels, Tb, Bun, lactate levels, PTT, Neut and RAR were independently associated with the 28-day mortality. We constructed four multivariate Cox regression models to assess the independent effect of RAR on the 28-day mortality rate in patients with AP (Table 2). The elevation of RAR was significantly correlated with an increased 28-day mortality rate both before (HR, 2.72; 95% CI, 1.64–4.52; $p < 0.001$) and after PSM (HR, 3.39; 95% CI, 1.76–6.51; $p < 0.001$).

## Linear relationship between the RAR and 28-day all-cause mortality

To further investigate the relationship between 28-day all-cause mortality and RAR, we employed adjusted restricted cubic spline regression analysis. This method allowed for the examination of potential non-linear relationships between RAR and mortality. As depicted in

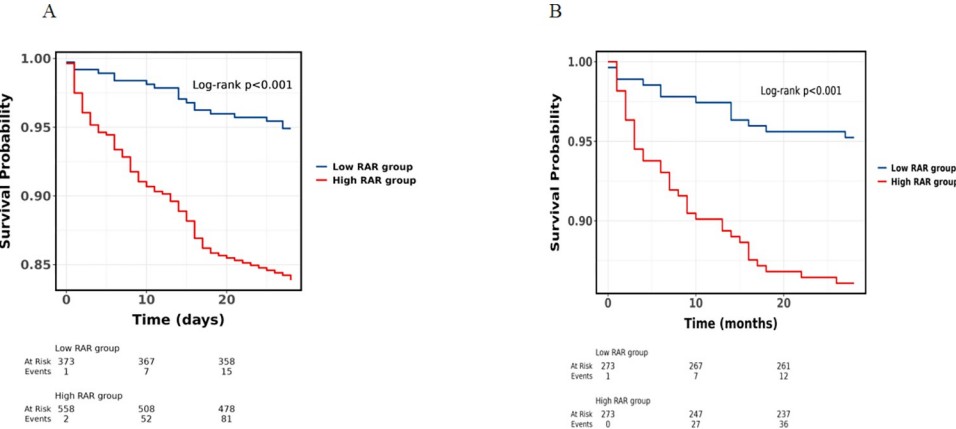

**Fig 2. Kaplan-Meier survival curves for 28-day mortality group by cutoff of RAR in unmatched and propensity score-matched cohorts.** (A) In the unmatched cohort, higher value of RAR was associated with deteriorated 28-day survival (HR, 3.37; 95%CI, 2.06–5.54; p<0.001). (B) In the propensity score-matched cohort, higher value of RAR was also associated with deteriorated 28-day survival (HR, 3.07; 95% CI, 1.64–5.77; p<0.001). Abbreviations: HR, hazard ratio; 95% CI, 95% confidence interval.

Fig 3, the analysis revealed a linear correlation between the RAR and the risk of 28-day all-cause mortality (p for overall = 0.003, p for nonlinear = 0.093). This indicated that as the RAR increased, the risk of mortality increased in a consistent manner, without any significant curvature or threshold effect in the relationship within the range of RAR observed in the study population.

## Receiver operating characteristic analysis

In order to further reinforce the relationship between RAR and the prognosis of AP, ROC analysis was also conducted (Fig 4). The results showed that the area under the curve (AUC) for RAR predicting all-cause mortality at 28 days was 0.669(95%CI: 0.617, 0.720, p<0.05). Additionally, when comparing RAR with the traditional scoring system SOFA, the results showed that the AUC for SOFA score was 0.564 (95% CI 0.512, 0.616, p = 0.03), which was lower than that for RAR. This indicated that RAR had predictive value for the prognosis of patients with AP.

**Table 2. Association of RAR and the risk of 28-day mortality.**

| Models | Before PSM | | | After PSM | | |
|---|---|---|---|---|---|---|
| | HR | 95%CI | P-value | HR | 95%CI | P-value |
| Models1 | 3.37 | (2.06–5.54) | < .001 | 3.07 | (1.64–5.77) | < .001 |
| Models2 | 3.27 | (1.99–5.36) | < .001 | 3.22 | (1.71–6.08) | < .001 |
| Models3 | 2.72 | (1.64–4.52) | < .001 | 3.39 | (1.76–6.51) | < .001 |
| Models4 | 2.72 | (1.64–4.52) | < .001 | 3.39 | (1.76–6.51) | < .001 |

Model1: Crude.

Model2: Adjust: Ethnicity, age, gender, comorbidities.

Model3: Adjust: Ethnicity, age, gender, comorbidities, potassium, Tb, Bun, lactate, PTT, Neut.

Model4: Adjust: Ethnicity, age, gender, comorbidities, potassium, Tb, Bun, lactate, PTT, Neut, mechanical ventilation.

Abbreviations: HR: Hazard Ratio, CI: Confidence Interval.

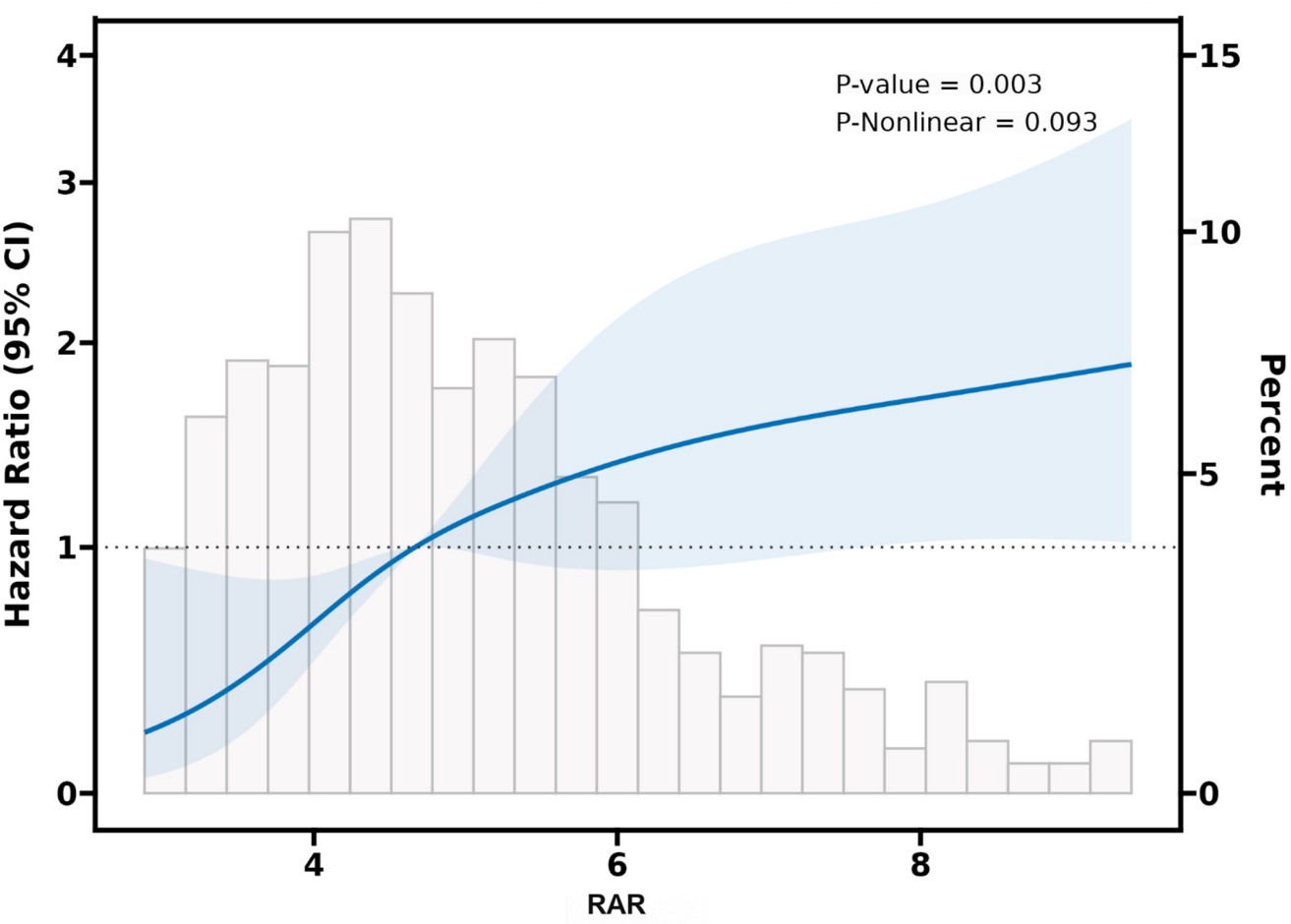

**Fig 3. Association between RAR and 28-day all-cause mortality with the RCS function.** Model with 4 knots located at 5th, 35th, 65th and 95th percentiles. Y-axis represents the HR to present survival for any value of RAR compared to individuals with reference value (50th percentile) of RAR.

## Subgroup analysis

Further research was performed on the relationship between RAR at admission and 28-day mortality in patients with AP, taking into consideration variables such as age, gender, comorbidity(include congestive heart failure, hypertension, chronic pulmonary disease, diabetes and renal failure), SAPSII, SOFA and mechanical ventilation (Fig 5), the critical value of SAPSII and SOFA was divided by median. We found that RAR showed no significant interaction effect in all subgroups. The result indicated that the predictive effect of RAR was consistent across different subgroups, and the association between RAR and mortality was robust, not influenced by the demographic and clinical variables studied.

## Sensitive analysis

To establish the robustness of the association between RAR and 28-day mortality in AP, in addition to adjusting the covariates in the COX model, we separately conducted COX regression analyses to calculate hazard ratios after: (1) excluding extreme values, and (2) stratifying patients into two groups based on the median RAR value. Outliers are defined as those observations that fall more than three standard deviations above or below the mean. The consistent

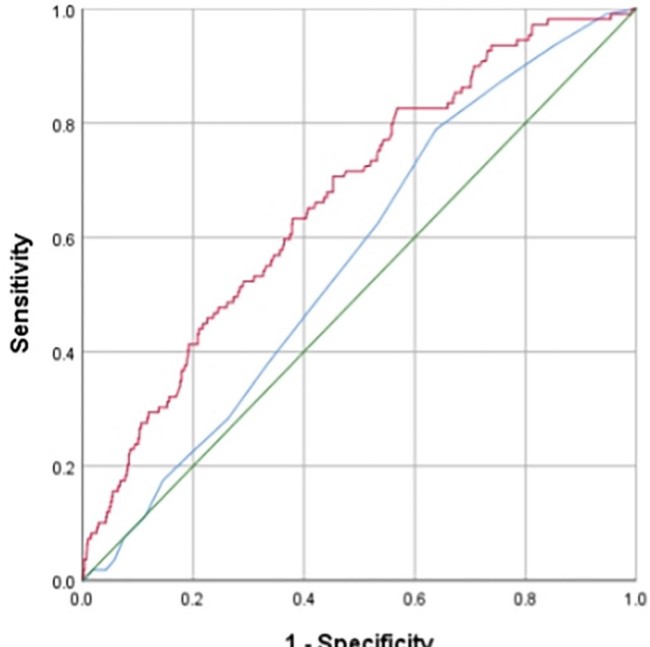

**Fig 4. ROC analysis of RAR for predicting 28-day mortality in acute pancreatitis.**

| Variable | Count | Percent | HR (95% CI) | P value | P for interaction |
|---|---|---|---|---|---|
| Overall | 931 | 100 | 3.38 (2.06 to 5.55) | <0.001 | |
| Age | | | | | 0.726 |
| > 65 | 331 | 35.6 | 3.84 (1.61 to 9.14) | 0.002 | |
| <= 65 | 600 | 64.4 | 3.17 (1.73 to 5.80) | <0.001 | |
| Gender | | | | | 0.345 |
| Male | 518 | 55.6 | 2.67 (1.37 to 5.19) | 0.004 | |
| Female | 413 | 44.4 | 4.37 (2.07 to 9.24) | <0.001 | |
| Congestive heart failure | | | | | 0.857 |
| No | 711 | 76.4 | 3.29 (1.84 to 5.86) | <0.001 | |
| Yes | 220 | 23.6 | 3.62 (1.39 to 9.44) | 0.008 | |
| Hypertension | | | | | 0.289 |
| No | 436 | 46.8 | 4.82 (2.04 to 11.41) | <0.001 | |
| Yes | 495 | 53.2 | 2.70 (1.47 to 4.95) | 0.001 | |
| Chronic pulmonary disease | | | | | 0.555 |
| No | 780 | 83.8 | 3.55 (2.08 to 6.07) | <0.001 | |
| Yes | 151 | 16.2 | 2.34 (0.63 to 8.63) | 0.203 | |
| Diabetes | | | | | 0.854 |
| No | 719 | 77.2 | 3.47 (1.91 to 6.32) | <0.001 | |
| Yes | 212 | 22.8 | 3.11 (1.29 to 7.51) | 0.012 | |
| Renal failure | | | | | 0.527 |
| No | 800 | 85.9 | 3.63 (2.09 to 6.32) | <0.001 | |
| Yes | 131 | 14.1 | 2.42 (0.80 to 7.36) | 0.119 | |
| Mechanial ventilation | | | | | 0.074 |
| No | 511 | 54.9 | 5.01 (2.49 to 10.09) | <0.001 | |
| Yes | 420 | 45.1 | 1.99 (0.97 to 4.08) | 0.061 | |
| SAPSII | | | | | 0.138 |
| > 36 | 448 | 48.1 | 2.35 (1.23 to 4.50) | 0.01 | |
| <= 36 | 483 | 51.9 | 5.12 (2.32 to 11.27) | <0.001 | |
| SOFA | | | | | 0.668 |
| > 5 | 387 | 41.6 | 2.87 (1.30 to 6.31) | 0.009 | |
| <= 5 | 544 | 58.4 | 3.59 (1.89 to 6.82) | <0.001 | |

**Fig 5. Subgroup analysis for 28-day mortality.**

findings from both the outlier-removed and median RAR-based patient grouping COX regression analyses underscored the robust association between the RAR and the 28-day mortality risk in AP. Despite the different analytical approaches, initially removing 10 outliers to yield an HR of 3.28 (95% CI 1.99 to 5.39, $p < 0.001$) and then reclassifying patients resulting in an HR of 2.60 (95% CI 1.72 to 3.95, $P < 0.001$), the statistical significance and the direction of the association remained unchanged. This provides strong evidence for the reliability of RAR as a prognostic marker for mortality in acute pancreatitis.

## Discussion

As a large, publicly available database, MIMIC-III has obtained increasing attention. Researchers determine the prognosis of critically ill patients based on their vital signs, auxiliary examinations, and relevant treatment measures at admission. Zhong et al.'s study based on MIMIC-III found that hemoglobin-to-RDW at admission was a significant prognostic marker for long-term mortality in patients with sepsis [12]. Research exploring the relationship between RAR and disease prognosis through the MIMIC database is not uncommon. Li et al.'s study found that RAR might be a potential biomarker for the prognostic assessment of acute myocardial infarction, and high RAR levels were associated with an increased risk of 90-day mortality in AMI patients [13]. In addition, RAR has been confirmed to be a risk factor for Aortic Aneurysms [14]. In the same measure, this study explored the association between RAR at admission and all-cause mortality in patients with AP. This study used propensity score matching to conduct Kaplan-Meier survival analysis before and after matching and constructed multivariate COX regression models. The results showed that an elevated RAR was significantly associated with an increased 28-day all-cause mortality rate in patients with AP and was an independent risk factor. The hazard ratios (HR) were 2.72 (95%CI, 1.64–4.52; p<0.001) before matching and 3.39(95%CI, 1.76–6.51; p<0.001) after matching. In addition, the linear relationship between RAR and the risk of death was confirmed by a restricted cubic spline plot, indicating that the higher the RAR was, the greater the risk of death for patients. This study also performed ROC curve analysis to assess the value of the RAR in predicting the 28-day all-cause mortality rate in patients with AP. The results revealed that the AUC for the RAR was 0.669(95%CI, 0.617–0.720; p<0.05), identifying the predictive value of RAR for the prognosis of AP. Finally, through the analysis of different subgroups, we found that RAR was associated with the 28-day all-cause mortality in patients with AP across subgroups including different genders, ages, comorbidities, SOFA scores, and SAPSII scores. This indicated that the predictive value of RAR could be broadly applicable and had good generalizability. The highlight of this study was existing research on the predictive value of RAR at admission for mortality in patients with AP was limited. This study obtained patients data from a large public database, employed a variety of statistical analysis methods to validate the relationship between RAR and the prognosis of patients with AP from different perspectives, ensuring the scientificity and rigor of the research findings.

When comparing our study with other earlier research, it is important to note that the study by Donmes et al. [15] revealed that there were indeed differences in RAR among patients with acute biliary pancreatitis of varying severities, with the highest RAR found in the severe acute pancreatitis group. Furthermore, they also found a dose-response relationship between RAR and mortality, which aligns with the findings of our study. Acehan et al. investigated the relationship between RAR and in-hospital mortality and severity in patients with AP. The results showed that RAR at 48 hours after admission was an independent predictor of SAP, with a cutoff value of 4.35. For in-hospital mortality, the AUC value of RAR-48th was 0.960 (95% CI: 0.931–0.989), significantly higher than the AUC values of existing scoring systems

[16]. Although the endpoint of this study was 28-day all-cause mortality and not in-hospital mortality, the endpoint time was relatively early. However, the results of this study showed that the AUC value of RAR at admission was 0.669, which is significantly different from the AUC value obtained by Acehan et al. The reason for the difference may be that previous studies have exaggerated the predictive value of RAR. In addition, the subjects of this study were AP patients hospitalized in the ICU, while previous studies included AP patients regardless of whether they were hospitalized in the ICU. Therefore, for patients with severe acute pancreatitis, the predictive value of RAR for prognosis may be limited. Although there is a significant difference in AUC, a similar cutoff value was found between this study and Acehan's study. Hidalgo et al. [17] found that the Charlson and Elixhauser comorbidity indices could be used to predict early mortality in patients with pancreatitis in a study involving 110,021 patients. The corresponding AUCs were 0.633 (95% CI 0.623–0.641) and 0.666 (95% CI 0.657–0.674), respectively. In this study, the RAR has an AUC value of 0.669 (95% CI: 0.617, 0.720), which is higher than the Charlson and Elixhauser comorbidity indices, and also superior to the AUC value of 0.651 (95% CI 0.597–0.701) for PNI reported by Efgan et al. in their research [18]. This suggests that RAR may be a more effective predictive marker for identifying individuals with acute pancreatitis who are at a higher risk of early mortality. Although the RAR has a relatively lower AUC compared to traditional scoring systems such as RANSON and Bedside Index of Severity in Acute Pancreatitis (BISAP) scoring systems [19], it demonstrates distinct advantages in terms of timeliness and cost-effectiveness.

Red blood cell distribution width is an automatically measured index of the heterogeneity of the erythrocytes. It is computed by dividing the standard deviation (SD) of RBC volume by MCV, and then multiplying by 100 to show the result as a percentage [20]. Initially, Şenol et al. discovered that RDW on admission was a predictor of mortality in patients with AP, with AUC 0.817(95%CI, 0.689–0.946; p<0.001) [21]. In subsequent studies, Amira et al. found that RDW was associated with the severity of patients with pancreatitis, RDW values were significantly higher in patients with bedside index for severity in acute pancreatitis (BISAP) scores≥3 (15.6%, 95%CI,14–16.9) compared to those with BISAP scores <3 (13.5%, 95%CI, 13–14.1) (p<0.001) [22]. The correlation between RDW alterations and the severity and prognosis of AP implies the occurrence of complex underlying mechanisms. RDW reflects the variability in the size of RBC in the circulation. It is routinely accounted by automated lab equipment used to conduct complete blood counts. RDW can contribute to diagnosis of certain types of anemias, particularly those that are caused by iron deficiency and those due to vitamin B12 or folic acid deficiencies. Severe blood loss leads to more immature cells being released into the circulate, abnormal hemoglobin or hemolysis can alter the shape of RBC additionally, resulting in increased RDW [23]. It could be conjectured that the connection between RDW and plasma inflammatory markers may be a secondary phenomenon stemming from underlying abnormal ferritin levels or anemia. Increased levels of inflammatory cytokines and variations in iron metabolism conditions related to inflammatory states may lead to the progression of anemia [24]. For instance, interferon and tumor necrosis factors can aggravate anemia by lessening colony formation of burst-forming unit erythroid cells and colony forming unit erythroid cells. The significant repressive effects of these inflammatory cytokines on erythroid progenitor cells may be primarily associated with their capability to decrease endothelial nitric oxide production, which is generally known to stimulate the proliferation of erythroid progenitor cells [25]. Through the aforementioned mechanisms, RDW can reflect the severity of systemic inflammatory response, which helps in assessing the prognosis of patients with pancreatitis.

ALB is produced in the liver and represents the most plentiful plasma protein. It has a variety of physiological functions, including maintaining osmotic pressure, protecting microvessels, reducing increased vascular permeability, binding endogenous and exogenous substances,

serving as an antioxidant and scavenging free radicals, exerting anticoagulant effects, maintaining acid-base balance, having anti-inflammatory effects, and preventing cell apoptosis [26]. It diminishes when facing low nutritional status and inflammation within the body [20]. Hong et al. found that a low serum albumin was independently associated with an increased risk of developing of persistent organ failure and death in acute pancreatitis, which might also be useful for the prediction of the severity of acute pancreatitis [27]. Moreover, hypoalbuminemia is frequently observed in critically ill patients. Compared with single levels of RDW or ALB, RAR can supply a thorougher reflection of clinical information for evaluating the patient's condition and prognosis. Wang's research showed that RAR had a higher AUC for predicting severe acute pancreatitis compared to RDW or albumin [28]. RAR was increasingly being applied in the assessment of disease severity and the prediction of disease prognosis. Hao et al.'s study indicated that a higher baseline RAR was associated with an increased risk of all-cause and cause-specific mortality in the general population, which suggested that RAR might be a simple, reliable, and inexpensive indicator for identifying individuals at high risk of mortality in clinical practice [29]. Chen et al. found that RAR might potentially become a valuable prognostic biomarker for patients with coronary heart disease and diabetes in the ICU. Elevated RAR levels are significantly associated with an increased risk of mortality during hospitalization [30].

Our study has the following advantages compared to other studies: by using PSM to balance the distribution of multiple covariates, we can more accurately estimate the impact of RAR on the prognosis of patients with AP. Simultaneously, this study improved the predictive ability of the model by constructing multivariate COX regression models and adjusting variables, thereby controlling for confounding factors. Finally, a robust relationship between RAR and the 28-day mortality in AP was established through subgroup analysis and sensitive analysis.

## Limitation

This study also has some limitations. First and foremost, as a retrospective study, inevitable biases exist in this study, which may affect the authenticity of the results. Secondly, owing to missing data in public databases, much of the information that could have affected the study was not collected, such as the results of blood gas analysis. In addition, the study did not include enough related information on treatment, which would allow for a more comprehensive evaluation of patient outcomes. Moreover, while this research identifies an association between RAR and adverse outcomes, it does not establish causality. Besides, when conducting subgroup analysis, it was not discussed whether there existed a relationship between RAR and mortality in patients with pancreatitis of different causes. Another limitation is reliance on a single database limits the generalizability of the results, therefore, it is essential to conduct prospective external validation of our results before future application. Finally, the AUC for RAR is 0.669, which means the predictive ability is limited, so the potential for combining RAR with other indicators to enhance predictive power should be explored.

## Conclusion

Our findings indicate a strong correlation between increased RAR and elevated overall mortality risk in patients with AP, suggesting that RAR may serve as an independent and significant indicator for assessing mortality risk in this population. This result needs to be confirmed by more prospective clinical studies.

## Supporting information

**S1 Table. Collinearity assessment of all variables.**
(PDF)

**S2 Table. Baseline characteristics of patients grouped by the survival status at 28 days.**
(PDF)

**S3 Table. Univariate and multivariable cox regression analysis of factors influencing 28-day mortality.**
(PDF)

## Author Contributions

**Conceptualization:** Lianyi Liao, Qingjun Deng.

**Data curation:** Qingsong Wu, Lianyi Liao.

**Formal analysis:** Qingsong Wu.

**Investigation:** Qingsong Wu.

**Methodology:** Qingsong Wu.

**Project administration:** Qingsong Wu.

**Resources:** Qingsong Wu, Lianyi Liao.

**Software:** Qingsong Wu, Lianyi Liao.

**Supervision:** Qingjun Deng.

**Validation:** Qingsong Wu.

**Visualization:** Qingsong Wu.

**Writing – original draft:** Qingsong Wu.

**Writing – review & editing:** Qingsong Wu.

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
