## [Decision Letter · Decision Letter 0]

26 Nov 2024

PONE-D-24-38811Association between red blood cell distribution width-to-albumin ratio at admission and all-cause mortality in patients with acute pancreatitis based on the MIMIC-III databasePLOS ONE

Dear Dr. Wu,

Thank you for submitting your manuscript to PLOS ONE. After careful consideration, we feel that it has merit but does not fully meet PLOS ONE’s publication criteria as it currently stands. Therefore, we invite you to submit a revised version of the manuscript that addresses the points raised during the review process.

We look forward to receiving your revised manuscript.

Kind regards,

Jian Wu, M.D, Ph.D

Academic Editor

PLOS ONE

Journal Requirements:

3. We notice that your supplementary figures are included in the manuscript file. Please remove them and upload them with the file type 'Supporting Information'. Please ensure that each Supporting Information file has a legend listed in the manuscript after the references list.

Reviewers' comments:

Reviewer's Responses to Questions

**Comments to the Author**

1. Is the manuscript technically sound, and do the data support the conclusions?

Reviewer #1: Yes

Reviewer #2: Yes

Reviewer #3: Yes

2. Has the statistical analysis been performed appropriately and rigorously? 

Reviewer #1: Yes

Reviewer #2: Yes

Reviewer #3: Yes

3. Have the authors made all data underlying the findings in their manuscript fully available?

Reviewer #1: Yes

Reviewer #2: Yes

Reviewer #3: Yes

4. Is the manuscript presented in an intelligible fashion and written in standard English?

Reviewer #1: Yes

Reviewer #2: Yes

Reviewer #3: Yes

5. Review Comments to the Author

Reviewer #1: The manuscript by Wu et al presents a retrospective analysis investigating the prognostic significance of the RAR in AP patients. The study leverages data from the MIMIC-III database and demonstrates a significant association between RAR levels and 28-day and 60-day all-cause mortality. The topic is relevant and interesting, given the ongoing efforts to identify more reliable biomarkers for predicting outcomes in AP patients. However, revisions are required, I recommend major revisions before this manuscript can be considered for publication. I would like to suggest some points for improvement.

Major points:

1. The quartile divisions for RAR seem arbitrary. Were alternative methods of stratification (e.g., tertiles, clinical thresholds) considered? If not, justify why quartiles were chosen.

2. Subgroup analysis results need further interpretation. The lack of significance in diabetic patients and those with high SAPSII scores should be discussed.

3. I recommend sensitivity analyses should also be performed to test the robustness of findings.

4. The RAR AUC values (0.669 and 0.671) suggest only modest discrimination. Please compare these values with other commonly used scoring systems or biomarkers in acute pancreatitis.

Minor points:

1. Expand the background on the clinical utility of RAR.

2. It will be better to highlight variables with significant differences in table.1.

3. I recommend that the author considers language polishing to improve clarity and readability. For example, replace "thoroughgoing estimation" with "detailed evaluation".

Reviewer #2: 1. Statistical and Methodological Issues

Handling of Missing Data:

The article mentions the use of multiple imputation to handle missing data but does not clearly describe the imputation methods and steps. It should be clarified whether appropriate imputation models (e.g., regression-based or machine learning methods) were used.

It is recommended to explicitly report the proportion and distribution of missing data, especially for key variables like RAR-related metrics, to determine if there is systematic missingness.

Rationale for Grouping:

RAR was divided into quartiles for grouping, which, while convenient, may obscure potential nonlinear associations. Was the sensitivity of the results to grouping strategies (e.g., logarithmic partitioning or other clinically relevant thresholds) evaluated?

Multicollinearity of Variables:

Although the article mentions evaluating multicollinearity, it does not clearly explain which variables were adjusted or excluded. It is recommended to further elaborate in the results section on how multicollinearity issues were addressed, especially for interrelated variables such as Hb, BUN, and PT.

2. Interpretation of Results and Logical Consistency

Kaplan-Meier Analysis:

The Kaplan-Meier curves show significant differences in 28-day and 60-day survival rates, but the lack of significant difference between Q2 and Q3 at 28 days (p = 0.366) is not fully explained. This could indicate minor biological differences between these groups and should be further discussed.

Cox Regression Model:

The Cox regression analysis identifies RAR as a significant risk factor but does not sufficiently explain the logic for selecting adjustment variables. If potential confounders were not included in the model, this might affect the conclusions.

The results section does not mention whether the proportional hazards assumption was tested. It should be clarified if this assumption was met.

3. Clinical Interpretation and Applicability

Clinical Significance:

Although RAR is statistically significant, its actual clinical relevance has not been adequately explored. For instance, does RAR significantly improve the predictive power of existing scoring systems like SOFA or APACHE-II? A comparative analysis using ROC curves is suggested.

Comparison with Existing Studies:

The article mentions differences between RAR and other indicators (e.g., CRP/lymphocyte ratio) but does not discuss RAR's specific advantages (e.g., cost, ease of measurement). Are there direct comparative data available?

4. Potential Language and Expression Issues

Ambiguous Statements:

For example: "A significant positive correlation was noticed between RAR quartiles and all-cause mortality (p < 0.001)".

This statement is overly broad. It is recommended to specify what "significant" means in terms of effect size or HR range.

Suggested revision: "RAR quartiles demonstrated a dose-response relationship with all-cause mortality, with a hazard ratio of up to X.X in the highest quartile (p < 0.001)."

Inconsistent Terminology:

Terms such as "RAR" are sometimes referred to as a ratio and at other times as an index. Consistency should be maintained throughout the article, and its definition should be clarified when first introduced.

5. Potential Data Issues

Baseline Characteristics Imbalance:

Some baseline characteristics (e.g., ALT, PTT) show significant differences across groups in Table 1, but the article does not discuss their potential influence. It is recommended to address the potential biases introduced by these differences.

Lack of External Validation:

Reliance on a single database (MIMIC-III) limits the generalizability of the results. Mentioning the need for future validation with independent datasets in the discussion would strengthen the study.

6. Technical Details

ROC Analysis:

The reported AUC of 0.669, though statistically significant, indicates limited predictive ability. This should be acknowledged in the discussion, and the potential for combining RAR with other indicators to enhance predictive power should be explored.

Adjustment of Variables:

The article mentions adjusting for all variables with p < 0.05, but the comprehensiveness and methodology of the adjustments are unclear. For example, was the role of gender and other potential confounders thoroughly addressed?

7. Formatting and Citation Issues

Text Formatting:

Ensure that the reference style fully complies with journal requirements.

Citation Count:

The number of references appears to be relatively low. Consider adding citations to fully support the content of the article.

Conclusion:

The overall conclusions of the article are reliable, but there is room for improvement in the transparency of methods, interpretation of results, and discussion of clinical relevance. Addressing the above issues can significantly enhance the scientific quality and impact of the article.

Reviewer #3: This is an interesting study. Based on the MIME-III database, the authors explored the relationship between red blood cell distribution width and albumin ratio and all-cause mortality in patients with acute pancreatitis upon admission. The perspective is relatively novel and holds significant clinical value. I have a few main questions:

1. What is the rationale for setting the age threshold at 58.3 years?

2. Is there any additional data to further validate the relevant results and conclusions?

6. PLOS authors have the option to publish the peer review history of their article (what does this mean?). If published, this will include your full peer review and any attached files.

Reviewer #1: No

Reviewer #2: No

Reviewer #3: No

---

## [Author Response · Author response to Decision Letter 0]

27 Dec 2024

1.Response to comment: The article mentions the use of multiple imputation to handle missing data but does not clearly describe the imputation methods and steps. It should be clarified whether appropriate imputation models (e.g., regression-based or machine learning methods) were used.

Response: We are very sorry for our negligence of unclearly describing the imputation methods and steps. After revision, specific methods and steps have been added to the article.We performed these operations by using the mice package in R software and carried out the imputation through regression-based models.

2.Response to comment: It is recommended to explicitly report the proportion and distribution of missing data, especially for key variables like RAR-related metrics, to determine if there is systematic missingness.

Response: After the data extraction was completed, a missing value analysis was conducted, and variables with more than 20% missing values were deleted, including pH, PCO2, PO2, and CRP. Additionally, cases with missing RAR, which is either RDW or ALB (a total of 3 cases), were excluded.

3.Response to comment: RAR was divided into quartiles for grouping, which, while convenient, may obscure potential nonlinear associations. Was the sensitivity of the results to grouping strategies (e.g., logarithmic partitioning or other clinically relevant thresholds) evaluated?

Response: The concern raised by the reviewers about the potential nonlinear associations by the quartile grouping of the RAR variable is one that we take very seriously. Here is our analysis and adjustment of this issue:The choice to use quartile grouping was based on its ability to simplify the interpretation of the data and to make it more accessible for clinicians to understand and apply. However, we fully acknowledge that this grouping strategy may fail to capture nonlinear relationships between variables. To address this, we have adjusted our grouping strategy. First, we conducted a ROC analysis to determine the cutoff value, and then we grouped the patients based on this cutoff value. The cutoff value for this study is 4.39, which is similar to that identified in previous study (4.35). Additionally, we utilized restricted cubic splines to explore the nonlinear relationship between RAR and survival time. The results indicated that there was no nonlinear relationship between RAR and survival time.

4.Response to comment: Although the article mentions evaluating multicollinearity, it does not clearly explain which variables were adjusted or excluded. It is recommended to further elaborate in the results section on how multicollinearity issues were addressed, especially for interrelated variables such as Hb, BUN, and PT.

Response: In the results section of the article, it is mentioned that the variance inflation factor method was used to assess multicollinearity among variables. As seen in the supplementary materials, the collinearity analysis results indicated that all variance inflation factors for the included variables were less than 10, suggesting that there was no multicollinearity among the variables. Therefore, no variable packages were excluded in the final analysis.

5.Response to comment: The Kaplan-Meier curves show significant differences in 28-day and 60-day survival rates, but the lack of significant difference between Q2 and Q3 at 28 days (p = 0.366) is not fully explained. This could indicate minor biological differences between these groups and should be further discussed.

Response: Due to the addition of Propensity Score Matching and Restricted Cubic Spline analyses, the content of the article has increased significantly. Therefore, after revision, the article focuses solely on the impact of RAR on the 28-day mortality of acute pancreatitis. The groups have been reduced from four to two. After revision, the Kaplan-Meier Analysis showed that the 28-day mortality of patients in the high RAR group was significantly increased compared to the low RAR group in both the unmatched cohort and the propensity score-matched cohort.

6.Response to comment: The Cox regression analysis identifies RAR as a significant risk factor but does not sufficiently explain the logic for selecting adjustment variables. If potential confounders were not included in the model, this might affect the conclusions.

Response: Regarding the selection of adjustment variables, in model 2, the adjustment variables are demographic characteristics. Model 3 builds upon Model 2 by adding variables with a p-value less than 0.05 in the multivariate analysis, including potassium, total bilirubin, blood urea nitrogen, lactate, partial thromboplastin time, and neutrophil count. Model 4 further includes the use of invasive mechanical ventilation on basis of the variables included in Model 3, in an effort to encompass as many potential confounders as possible.

7.Response to comment: The results section does not mention whether the proportional hazards assumption was tested. It should be clarified if this assumption was met.

Response: Before conducting the Cox regression analysis, the proportional hazards assumption was assessed using the Schoenfeld residuals method. The results indicated that all independent variables satisfied the proportional hazards assumption. The aforementioned content has been added to study outcomes section in the article.

8.Response to comment: Although RAR is statistically significant, its actual clinical relevance has not been adequately explored. For instance, does RAR significantly improve the predictive power of existing scoring systems like SOFA or APACHE-II? A comparative analysis using ROC curves is suggested.

Response: The data for this study were sourced from the MIMIC-III database, which didn’t include APACHE-II scores. This study found that the AUC of RAR was higher than that of SOFA, indicating that RAR had predictive value for the prognosis of patients with acute pancreatitis.

9.Response to comment: The article mentions differences between RAR and other indicators (e.g., CRP/lymphocyte ratio) but does not discuss RAR's specific advantages (e.g., cost, ease of measurement). Are there direct comparative data available?

Response: After reviewing the literature, there are no detailed data directly comparing RAR with other indicators in terms of cost, ease of measurement, and other specific advantages. Most studies focus on the clinical utility and value of these indicators in predicting the prognosis of specific diseases. However, RAR is an easily obtainable and inexpensive indicator.

10.Response to comment: Ambiguous Statements and inconsistent terminology

Response: After revision, the expression in parts of the article has been adjusted, and terms have been modified to maintain consistency.

11.Response to comment: Some baseline characteristics (e.g., ALT, PTT) show significant differences across groups in Table 1, but the article does not discuss their potential influence. It is recommended to address the potential biases introduced by these differences.

Response: We used the Propensity Score Matching method to reduce the imbalance of baseline characteristics between groups, thereby avoiding the impact of these imbalances on the outcomes. We conducted Kaplan-Meier analysis and constructed COX regression models using data from before and after PSM, respectively, and the results from the two groups were consistent.

12.Response to comment: Reliance on a single database (MIMIC-III) limits the generalizability of the results. Mentioning the need for future validation with independent datasets in the discussion would strengthen the study.

Response: The data in this study is solely derived from the MIMIC III database, which limits the generalizability of the results. As Reviewer suggested, in the limitations section of the discussion, we mentioned that the necessity of validating the findings using other datasets in the future is required to strengthen the results of this study.

13.Response to comment: The reported AUC of 0.669, though statistically significant, indicates limited predictive ability. This should be acknowledged in the discussion, and the potential for combining RAR with other indicators to enhance predictive power should be explored.

Response: In this study, the AUC of RAR is 0.669, indicating limited predictive power. Considering the Reviewer' s suggestion, in the discussion, we have stated that the potential for combining RAR with other indicators to enhance predictive power should be explored.

14.Response to comment: The article mentions adjusting for all variables with p < 0.05, but the comprehensiveness and methodology of the adjustments are unclear. For example, was the role of gender and other potential confounders thoroughly addressed?

Response: Regarding the selection of adjustment variables, in Model 2, the adjustment variables consist of demographic characteristics, including gender, age, race, and comorbidities. Model 3 builds upon Model 2 by adding variables with a p-value less than 0.05 in the multivariate analysis, which includes potassium, total bilirubin, blood urea nitrogen, lactate, partial thromboplastin time, and neutrophil count. Model 4 further includes the use of invasive mechanical ventilation on basis of the variables included in Model 3, in an effort to control for as many potential confounding factors as possible.

15.Response to comment: Ensure that the reference style fully complies with journal requirements.

Response:The reference style has been adjusted according to the requirements of the journal.

16.Response to comment: The number of references appears to be relatively low. Consider adding citations to fully support the content of the article.

Response: The number of references has been appropriately increased to support the content of the article.

---

## [Decision Letter · Decision Letter 1]

10 Jan 2025

PONE-D-24-38811R1Association between red blood cell distribution width-to-albumin ratio at admission and all-cause mortality in patients with acute pancreatitis based on the MIMIC-III databasePLOS ONE

Dear Dr. Wu,

Thank you for submitting your manuscript to PLOS ONE. After careful consideration, we feel that it has merit but does not fully meet PLOS ONE’s publication criteria as it currently stands. Therefore, we invite you to submit a revised version of the manuscript that addresses the points raised during the review process.

We look forward to receiving your revised manuscript.

Kind regards,

Jian Wu, M.D, Ph.D

Academic Editor

PLOS ONE

Reviewers' comments:

Reviewer's Responses to Questions

**Comments to the Author**

1. If the authors have adequately addressed your comments raised in a previous round of review and you feel that this manuscript is now acceptable for publication, you may indicate that here to bypass the “Comments to the Author” section, enter your conflict of interest statement in the “Confidential to Editor” section, and submit your "Accept" recommendation.

Reviewer #1: (No Response)

Reviewer #2: All comments have been addressed

Reviewer #3: All comments have been addressed

2. Is the manuscript technically sound, and do the data support the conclusions?

Reviewer #1: Yes

Reviewer #2: Yes

Reviewer #3: Yes

3. Has the statistical analysis been performed appropriately and rigorously? 

Reviewer #1: Yes

Reviewer #2: Yes

Reviewer #3: Yes

4. Have the authors made all data underlying the findings in their manuscript fully available?

Reviewer #1: Yes

Reviewer #2: Yes

Reviewer #3: Yes

5. Is the manuscript presented in an intelligible fashion and written in standard English?

Reviewer #1: Yes

Reviewer #2: Yes

Reviewer #3: Yes

6. Review Comments to the Author

Reviewer #1: It seems that the authors did not address my comments. Could it be that they were overlooked? I am reiterating my previous comments below and kindly ask the authors to check the decision letters.

The manuscript by Wu et al presents a retrospective analysis investigating the prognostic significance of the RAR in AP patients. The study leverages data from the MIMIC-III database and demonstrates a significant association between RAR levels and 28-day and 60-day all-cause mortality. The topic is relevant and interesting, given the ongoing efforts to identify more reliable biomarkers for predicting outcomes in AP patients. However, revisions are required, I recommend major revisions before this manuscript can be considered for publication. I would like to suggest some points for improvement.

Major points:

1. The quartile divisions for RAR seem arbitrary. Were alternative methods of stratification (e.g., tertiles, clinical thresholds) considered? If not, justify why quartiles were chosen.

2. Subgroup analysis results need further interpretation. The lack of significance in diabetic patients and those with high SAPSII scores should be discussed.

3. I recommend sensitivity analyses should also be performed to test the robustness of findings.

4. The RAR AUC values (0.669 and 0.671) suggest only modest discrimination. Please compare these values with other commonly used scoring systems or biomarkers in acute pancreatitis.

Minor points:

1. Expand the background on the clinical utility of RAR.

2. It will be better to highlight variables with significant differences in table.1.

3. I recommend that the author considers language polishing to improve clarity and readability. For example, replace "thoroughgoing estimation" with "detailed evaluation".

Reviewer #2: (No Response)

Reviewer #3: The author has made appropriate modifications in response to the reviewer’s suggestions and has addressed the reviewer’s comments. Currently, I have no further comments, and I believe the current version is suitable for publication in the journal PLOS ONE.

7. PLOS authors have the option to publish the peer review history of their article (what does this mean?). If published, this will include your full peer review and any attached files.

Reviewer #1: No

Reviewer #2: No

Reviewer #3: No

---

## [Author Response · Author response to Decision Letter 1]

14 Jan 2025

1.Response to comment: The quartile divisions for RAR seem arbitrary. Were alternative methods of stratification (e.g., tertiles, clinical thresholds) considered? If not, justify why quartiles were chosen.

Response: I acknowledge that the issue of arbitrary grouping based on quartiles was identified during the initial revision of the article. Initially, the choice to group the data by quartiles was made, which indeed may have seemed somewhat arbitrary. Reflecting on this, I have made a crucial adjustment to the grouping criteria. For this study, I have now adopted the cutoff value of RAR as the grouping criterion, which aligns more closely with the clinical thresholds established in previous research. This change aims to provide a more clinically relevant and justified basis for the groupings in our analysis.

2.Response to comment: Subgroup analysis results need further interpretation. The lack of significance in diabetic patients and those with high SAPSII scores should be discussed.

Response: Following the regrouping based on the cutoff value of RAR, a subgroup analysis was conducted to explore whether the association between RAR and 28-day mortality varied across different subgroups of patients. The results of this analysis indicated that the RAR did not demonstrate significant interaction effects across all the subgroups examined. This suggests that the relationship between RAR and 28-day mortality is consistent across various patient characteristics.

3.Response to comment: I recommend sensitivity analyses should also be performed to test the robustness of findings.

Response: To establish the robustness of the association between RAR and 28-day mortality in AP, in addition to adjusting the covariates in the COX model, we separately conducted COX regression analyses to calculate hazard ratios after: (1) excluding extreme values, and (2) stratifying patients into two groups based on the median RAR value. The consistent findings from both the outlier-removed and median RAR-based patient grouping COX regression analyses underscored the robust association between the RAR and the 28-day mortality risk in AP. 

4.Response to comment: Expand the background on the clinical utility of RAR.

Response: In the discussion part, we expanded the background of the clinical application of RAR. Studies have shown that higher baseline RAR is related to the increased risk of all-cause and cause specific mortality in the general population. In addition, RAR may become a valuable prognostic biomarker for patients with coronary heart disease and diabetes in the ICU.

5.Response to comment: It will be better to highlight variables with significant differences in table.1.

Response: The variables with significant differences have been highlighted in Table 1.

6.Response to comment: I recommend that the author considers language polishing to improve clarity and readability. For example, replace "thoroughgoing estimation" with "detailed evaluation".

Response: We agree that clear and precise language is crucial for effective communication of our research findings.In response to your suggestion, we have undertaken a comprehensive language polish throughout the manuscript to ensure that the text is both clear and readable.

---

## [Decision Letter · Decision Letter 2]

23 Jan 2025

Association between red blood cell distribution width-to-albumin ratio at admission and all-cause mortality in patients with acute pancreatitis based on the MIMIC-III database

PONE-D-24-38811R2

Dear Dr. Wu,

We’re pleased to inform you that your manuscript has been judged scientifically suitable for publication and will be formally accepted for publication once it meets all outstanding technical requirements.

Kind regards,

Jian Wu, M.D, Ph.D

Academic Editor

PLOS ONE

Additional Editor Comments (optional):

Reviewers' comments:

Reviewer's Responses to Questions

**Comments to the Author**

1. If the authors have adequately addressed your comments raised in a previous round of review and you feel that this manuscript is now acceptable for publication, you may indicate that here to bypass the “Comments to the Author” section, enter your conflict of interest statement in the “Confidential to Editor” section, and submit your "Accept" recommendation.

Reviewer #1: All comments have been addressed

Reviewer #2: All comments have been addressed

2. Is the manuscript technically sound, and do the data support the conclusions?

Reviewer #1: Yes

Reviewer #2: Yes

3. Has the statistical analysis been performed appropriately and rigorously? 

Reviewer #1: Yes

Reviewer #2: Yes

4. Have the authors made all data underlying the findings in their manuscript fully available?

Reviewer #1: Yes

Reviewer #2: Yes

5. Is the manuscript presented in an intelligible fashion and written in standard English?

Reviewer #1: Yes

Reviewer #2: Yes

6. Review Comments to the Author

Reviewer #1: The authors have addressed all the concerns and suggestions. The current version of the manuscript meets the standards for publication.

Reviewer #2: (No Response)

7. PLOS authors have the option to publish the peer review history of their article (what does this mean?). If published, this will include your full peer review and any attached files.

Reviewer #1: No

Reviewer #2: No

---

## [Editor Report · Acceptance letter]

30 Jan 2025

PONE-D-24-38811R2 

PLOS ONE

Dear Dr. Wu, 

I'm pleased to inform you that your manuscript has been deemed suitable for publication in PLOS ONE. Congratulations! Your manuscript is now being handed over to our production team.

Kind regards, 

on behalf of

Dr. Jian Wu 

Academic Editor

PLOS ONE